# Effect of Solvent on Superhydrophobicity Behavior of Tiles Coated with Epoxy/PDMS/SS

**DOI:** 10.3390/polym14122406

**Published:** 2022-06-14

**Authors:** Srimala Sreekantan, Ang Xue Yong, Norfatehah Basiron, Fauziah Ahmad, Fatimah De’nan

**Affiliations:** 1School of Materials and Mineral Resources Engineering, Engineering Campus, Universiti Sains Malaysia, Nibong Tebal 14300, Pulau Pinang, Malaysia; xueyong95@hotmail.com (A.X.Y.); fatehahbasiron31@gmail.com (N.B.); 2School of Civil Engineering, Engineering Campus, Universiti Sains Malaysia, Nibong Tebal 14300, Pulau Pinang, Malaysia; cefatimah@usm.my

**Keywords:** superhydrophobic, spray-coating, polydimethylsiloxane, silica solution, epoxy resin, acetone, hexane, isopropanol

## Abstract

Superhydrophobic coatings are widely applied in various applications due to their water-repelling characteristics. However, producing a durable superhydrophobic coating with less harmful low surface materials and solvents remains a challenge. Therefore, the aim of this work is to study the effects of three different solvents in preparing a durable and less toxic superhydrophobic coating containing polydimethylsiloxane (PDMS), silica solution (SS), and epoxy resin (DGEBA). A simple sol-gel method was used to prepare a superhydrophobic coating, and a spray-coating technique was employed to apply the superhydrophobic coating on tile substrates. The coated tile substrates were characterized for water contact angle (WCA) and tilting angle (TA) measurements, Field-Emission Scanning Electron Microscopy (FESEM), Atomic Force Microscopy (AFM), and Fourier Transform Infrared Spectroscopy (FTIR). Among 3 types of solvent (acetone, hexane, and isopropanol), a tile sample coated with isopropanol-added solution acquires the highest water contact angle of 152 ± 2° with a tilting angle of 7 ± 2° and a surface roughness of 21.80 nm after UV curing for 24 h. The peel off test showed very good adherence of the isopropanol-added solution coating on tiles. A mechanism for reactions that occur in the best optimized solvent is proposed.

## 1. Introduction

A material is known to be superhydrophobic when it has a water contact angle of more than 150° and a negligible titling angle of less than 10° [1,2]. Superhydrophobic coating has gained more attention and lead to many applications such as anti-icing [3,4], anti-fogging [5,6], oil-water separation [7,8], anti-corrosion [9,10,11], self-cleaning [12,13], antibacterial [14,15] and biomedical applications [16]. Various methods that have been used for preparation of rough surfaces are layer by layer assembly [17,18], spray-coating [19,20,21], lithography [22,23], sol gel processing [24,25], electrochemical deposition [26,27] and chemical vapour deposition [28]. Among them, spray-coating is a fabrication process used for industrial applications due to its availability in commercial form and a simple procedure that uses inexpensive materials [2,29]. In regard to this, the increase in surface roughness and the reduction of surface energy are vital in forming a superhydrophobic surface [30,31,32].

A summary of findings on the choice of materials used for spray-coating is shown in Table 1. As seen, the main precursors needed for superhydrophobic spraying techniques are solvent, low surface energy material and nanoparticles. The function of low surface energy material is to reduce the wettability of the surface up to 120°. Nanoparticles are to achieve appropriate roughness to trap the air to further reduce the wettability to achieve a water contact angle greater than 150° and a tilting angle smaller than 10°. The function of solvent is to reduce the viscosity of low surface energy material for easy application of superhydrophobic coating [33] on material surface. The emphasis of this work was on the solvent utilized to generate the superhydrophobic coating, as there have been many studies on nanoparticles and low surface energy materials.

According to Luo et al. the dispersibility of nanoparticles in solvent influences the generation of even dispersion or aggregation, which controls the formation of a smooth or rough coating on the surface of a substrate [45]. To produce a well-dispersed or aggregated superhydrophobic coating, the polarity and relative permittivity of the solvent are the important criteria to be considered. For instance, inorganic components such as SiO_2_ nanoparticles aggregate when non-polar solvents are used but disperse evenly in polar solvents. This happens because polar solvents stabilize silica dispersion through strong hydrogen bonding to silanol groups on the silica surface. Conversely, low-polarity solvents result in destabilization and gelation of silica particles via hydrogen bonding between adjacent silica particles [46,47]. The relative permittivity of the solvent is also a measure of solvent polarity; the lower the relative permittivity of the solvent, the lower the polarity of the mixture; the polarity of the mixture may eventually decrease to the point where it is no longer sufficient to sustain the dispersion of the polar silica nanoparticles, resulting in particle aggregation. Nonetheless, high aggregation conditions prior to the phase separation limit (gelation of SiO_2_ nanoparticles) are desirable as they help to create multi-scale roughness [47]. Therefore, an appropriate solvent needs to be selected to form the required roughness to reduce the wettability behavior.

A variety of solvents used by researchers to prepare superhydrophobic coating is tabulated in Table 2. In 2018, Zhang et al. used xylene as solvent for superhydrophobic epoxy/PDMS nanocomposite coating fabrication [6]. Tetrahydrofuran was also reported as solvent to synthesize superhydrophobic wood surfaces, hydrophobic sol-gel coating and UV-cured superhydrophobic cotton fabric surfaces [25,48,49]. Saleem used toluene as solvent in the development of superhydrophobic surfaces [33]. Among these solvents, hexane is the most popular solvent used to produce superhydrophobic coating due to low permittivity nature as it helps to create multi-scale roughness surface to trap air and increase the WCA. [12,35,50,51,52] are among the author researchers fabricated superhydrophobic coating by using hexane as solvent. However, these solvents are toxic and hazardous towards organs through prolonged exposure [53]. Other solvents such as isopropanol and acetone that has less hazardous effect towards users are not explored probably due to the relatively high permittivity value, 17.9 and 20.7, respectively as compared to hexane (1.9) 54. Due to safety concern, these two solvent are still explored in this work because the use of hydroxyl terminated PDMS (ε = 2.30 − 2.80) would react with mild polar hydroxyl and carbonyl group of those solvents because of the like-dissolve-like concept [54,55]. This would sustain the dispersion of the polar silica nanoparticles, with certain degree of aggregation. Besides, to reduce the toxicity of the superhydrophobic coating produced in this work, a low surface energy material, PDMS was used. To value add, the SiO_2_ nanoparticles that is required to increase the surface roughness for the superhydrophobic coating was extracted from palm oil fuel ash waste (POFA). A suitable mechanism was also proposed for the development of super-hydrophobic coating on the tiles.

## 2. Materials and Methods

### 2.1. Materials

In this work, the tiles with dimension size of 3 cm × 3 cm were used as substrate were obtained from Ceramic Research Company Sdn Bhd. Palm Oil Fuel Ash (POFA) and Hydroxyl-terminated Polydimethylsiloxane (PDMS-OH) act as precursor were obtained from Malpom Industries Berhad (Pulau Pinang, Malaysia) and Sigma Aldrich (Darmstadt, Germany), respectively. Sodium Hydroxide (NaOH) and Sulfuric Acid (H_2_SO_4_) with purity of 98% were used as POFA extraction and also Hexane (purity of 98.5%) and Isopropanol (purity of 99.8%) was used as solvent were purchased from Merck (Darmstadt, Germany). For solvent and cleaning agent, acetone; with purity of 99.9% was obtained from J.T. Baker (Avantor, Radnor, PA, USA). Meanwhile, epoxy resin crystal clear (diglycidyl ether of bisphenol A) were used as binder and epoxy hardener crystal clear (trimethyl hexamethylene diamine) were used as curing agent. Both materials were obtained from Euro Chemo-Pharma (Pulau Pinang, Malaysia). For the cross-linking agent, 3-aminopropyltrimethoxysilane (AMPS) with purity of 98% and dibutyltin dilaurate (DBTL) with purity of 98% was used as catalyst. Both materials were obtained from Sigma Aldrich, (Darmstadt, Germany). All the chemicals used were used as received without further purification.

### 2.2. Extraction of SS from POFA

Alkali extraction method was used to extract SS from POFA by mixing 10 g of POFA and 100 mL of 1 M NaOH at 80 °C for 1 h. After that, the mixture was allowed to cool to room temperature and filtered using a Whatman filter paper with pore size of 11 µm. Then, the filtrate was titrated against 1 M H_2_SO_4_, to adjust its pH to 3 and the solution produced was silica solution (SS) [36].

### 2.3. Preparation of the Hydrophobic Solution

The procedure of epoxy/PDMS/SS preparation is shown in Figure 1 and mainly consists of 3 mixtures (mixture A, B and C). Mixture A was prepared by stirring 100 mL of solvent and 10 mL of SS solution vigorously for 1 h. Next, 50 mL of solvent, 10 mL of PDMS, 5 mL of AMPS, and 1 mL of DBTL were stirred moderately and heated at 60 °C for 20 min to obtain Mixture B. Then, mixtures A and B were mixed and stirred vigorously for 1 h. Simultaneously, Mixture C was prepared by mixing 2 mL of epoxy resin with 1 mL of hardener. After that, mixture C was added into mixture A + B and stirred vigorously for 2 h. The same procedure is repeated for different types of solvent (acetone, hexane, isopropanol).

### 2.4. Fabrication of Hydrophobic Coating on Tiles Substrate

For the fabrication of film on tiles, the epoxy/PDMS/SS coating was sprayed on tile from 20 cm away with the aid of a spraying gun (air pressure: 40 psi), followed by 5 min of drying at 80 °C in an oven. The spraying and drying processes were repeated subsequently until a 3-layer Epoxy/PDMS/SS coating was obtained, and the coated substrate was cured overnight at 80 °C in an oven [36]. Super hydrophobicity characterization (WCA and TA) was used to analyze the coated substrate, and the best solvent was fixed for the Epoxy/PDMS/SS coating preparation for the rest of the experiment. Samples that were coated with acetone, hexane, and isopropanol-added solutions were labelled as S1, S2 and S3. The effect of UV curing was also studied on the respective samples and labelled as S1-UV, S2-UV and S3-UV. 

### 2.5. Characterizations and Analysis

#### 2.5.1. Water Contact Angle (WCA) and Tilting Angle (TA) Measurement

In this work, water contact angle and tilting angle of the samples were measured by a contact angle goniometer (Model 250-F1, Rame-Hart Instruments Co., Mountain Lakes, NJ, USA). The measurement was carried out by placing the samples in the goniometer that is attached to an Image analyzer. A drop of water with a volume of 5 µm was used to determine the water contact angle. Each sample was subjected to 10 measurements in each 4-angle position, including vertical left, vertical right, horizontal left, horizontal right. The WCA and TA were obtained by using DROPimage Advanced software [36]. The measurements were also carried out on the samples before and after durability test to investigate the super hydrophobicity of the sample. In addition, surface energy of coated samples was determined for a more thorough analysis on water contact angle measurement.

#### 2.5.2. Atomic Force Microscopy (AFM)

The surface topology was characterized using an atomic force microscope (AFM, Nano Navi, SPA400, Seiko Instruments, Chiba, Japan) operated in contact mode. The surface roughness of samples was measured, to study the effect of different solvent in PDMS/SS coating on water contact angle. Root-mean-square roughness of the samples was determined by operating at the contact mode of 5 µm × 5 µm. Surface roughness of the samples was analyzed by using NanoNavi software [58]. AFM images subjected to polynomial background subtraction.

#### 2.5.3. Field-Emission Scanning Electron Microscopy (FESEM)

Field-Emission Scanning Electron Microscope (FESEM-EDX, Supra 35VP, Zeiss, Oberkochen, Germany) was used to study the surface morphology of samples at an acceleration voltage of 5 kV. As the substrate was non-conductive, a thin layer of gold was sputtered onto the sample surface to make it conductive in order to obtain a clear FESEM image of the surface morphology [59].

#### 2.5.4. Fourier Transform Infrared Spectroscopy (FTIR)

In this work, transmission, and absorbance mode Fourier Transform Infrared (FTIR, Perkin Elmer, Ohio, United States) was used to investigate the functional groups present on the coated samples surface and the interface between coating and substrate, in order to study the formation of bonding after the application of PDMS/SS coating. Besides, the samples before and after immersion test were analyzed by using FTIR to evaluate the changes in the functional groups of the sample. The samples were tested in absorbance and transmittance mode from 4000 to 550 cm^−1^ wavenumber [35].

#### 2.5.5. Peel-Off Test

Peel-off test was carried out to investigate adhesion between PDMS/SS coating and substrate surface. It was carried out by using double-sided foam tape in which the tape was adhered to the surface of coating and pressed to confirm the adhesion was tight where no gap was found at the interface of tape and sample. Lastly, the tape was peeled off. This test was repeated for 5 cycles. The value of water contact angle before and after peel-off test was measured to evaluate the durability of the coating [28].

## 3. Results and Discussion

### 3.1. Effect of Solvent on Wettability

Table 3 shows the water contact angle, tilting angle, surface energy and the roughness of tiles coated with S1, S2 and S3 that were prepared with different solvents. Water contact angles for S1 and S2 coated with acetone and hexane added solution are 88 ± 1° and 85° ± 1°, respectively. Therefore, S1 and S2 exhibit hydrophilic behavior. On the other hand, S3 that was coated with isopropanol-added solution has water contact angles of 149 ± 2°, showing hydrophobic behavior. The high surface energy of the S1 and S2 coatings, which is 15 times higher than the S3 with 2 J/m^2^, contributes for the difference in water contact angle. S1 and S2 may indeed be affected by the dispersion of PDMS and SS particles in different solvents, which will be explained in more detail later. Furthermore, as compared to S1 and S2, sample S3 has a higher roughness, which could be another aspect that improves the sample’s hydrophobicity. As for tilting angle, it was found S1 with 7.09 nm has a lower tilting angle (20°) as compared to S2 that has a roughness of 13.01 with a tilting angle of 46°. The findings are consistent with [29,60], which demonstrated that for hydrophilic samples, higher surface roughness results in a higher tilting angle. Besides, higher surface roughness will also result in a stronger water pining effect due to the absence of air pockets. This situation will result in the penetration of water droplets into the grooves. Thus, S1 and S2 are predicted to be in a Wenzel state, as the Wenzel equation states that roughness emphasizes the effect of surface chemistry. In other words, for hydrophilic surfaces, the higher the surface roughness, the more hydrophilic the surface is while for a hydrophobic surface, the higher the surface roughness, the more hydrophobic the surface will be. The wetting mechanisms of hydrophobic surfaces (S3) can also be determined based on their respective tilting angles. The tilting angle of 10 ± 2° for S3 suggests the surface is in a Cassie-Baxter state, which is further supported by the surface morphology and high surface roughness (18.57 nm), which render a hierarchical structure. Such morphology leads to the formation of air voids that help in water droplet suspension, which results in a low tilting angle.

Apart from that, slight changes in water contact angle were also observed after the samples were cured under UV for 24 h (Table 3). After UV-curing, the water contact angles of samples that were coated with hexane-added solution and isopropanol-added solution were increased by ~3°, while the water contact angle of sample that was coated with acetone-added solution was decreased by ~6°. The increase in water contact angle of S2-UV and S3-UV after UV curing may be attributed to the increase in grafting density of the PDMS polymer chain to form a stronger 3D network 36. Besides, UV-curing helps to cure the polymer phase that was disrupted by aggregated particles in oven curing, producing a coating with strong adhesion, mechanical reliability, and chemical resistance [51,61,62]. Even though the difference in WCA was not statistically significant, the effect of UV was still considered crucial as the WCA was increased to above 150°, causing S3-UV to be superhydrophobic. However, the decrease in water contact angle of S1-UV after UV curing may be attributed to surface oxidation, which is triggered by the functional carbonyl group (C=O) presence in acetone. This accelerates the degradation of long alkyl chains into smaller chains [63,64], thus increasing the wettability.

### 3.2. Effect of Solvent on Surface Morphology and Surface Roughness

Surface roughness is another key contributor to the superhydrophobic characteristics of coated samples other than surface energy 35. Figure 1 shows the 3D AFM topographical images (Figure 1a–f), AFM line profile (Figure 1(a^a^–f^a^), Figure 1(c^c^,f^c^)) and FESEM images (Figure 1(a^b^–f^b^)) of S1, S2 and S3 before and after UV treatment. From Figure 1(a^b^,b^b^), it is observed that the surfaces of S1 and S2 before UV curing do not have obvious micro papillae structure. However, under AFM, the surface possesses certain roughness as indicated by the peaks and valleys in (Figure 1(a^a^)) and (Figure 1(b^a^)) corresponding to RMS of 7.09 nm and 13.01 nm, respectively. In comparison to S1 and S2, FESEM images of S3 before UV curing show that the surfaces are relatively rough (Figure 1(c^b^)). S3 that was coated with isopropanol-added solution has an RMS value of 18.57 nm. The high surface roughness in S3 is due to the dispersion of the coating solution, in which multiscale roughness is created. This sample has an appropriate degree of particle aggregation in the isopropanol-added solution, which induces hierarchical structure [47].

For samples cured with UV (Figure 1(d^b^,e^b^,f^b^)), a similar trend was observed as before UV curing (Figure 1(a^b^,b^b^,c^b^)), respectively. However, the distance between the hills for S1-UV after UV curing (Figure 1(d^a^)) is wider than that before UV curing (Figure 1(a^a^)), leading to a lower water contact angle after UV curing. Figure 1(e^a^) shows that S2-UV after UV curing has peaks and valleys with a lower contrast as compared to after UV curing, leading to a lower RMS roughness value of 11.57 nm. The S3-UV that was coated with isopropanol-added solution and UV-cured has the highest RMS roughness value of 21.80 nm. This is probably ascribed to the even distribution of peaks and valleys with spikes that renders a hierarchical structure in S3 after UV curing (Figure 1(f^a^,f^c^)) as compared to before curing (Figure 1(c^a^,c^c^)). In summary, the formation of hierarchical structures with appropriate roughness and distance between hills improves hydrophobicity. Those characteristics were achieved in the isopropanol-added solution that has been UV cured.

### 3.3. Effect of Solvent on Dispersion of Coating Solution

The effects of solvents such as hexane, acetone, and isopropanol on the dispersion of superhydrophobic coating solutions were investigated. Figure 2 shows that agglomeration was evident in all three solutions, but that after shaking, the isopropanol-added solution became well-dispersed whereas the agglomeration in the other two solutions remained. The relative permittivity of the solvent utilized causes such circumstance. Hexane is a non-polar solvent with a relative permittivity of 1.90, making it the least polar. As a result, it failed to disperse SS particles, which are polar in nature with OH groups. Hexane’s relative permittivity is too low, resulting in SS particle gelation, rather than particle aggregation. It was proven by the fact that samples S2 coated with hexane-added solution have low water contact angles before and after UV curing due to the absence of hierarchical structure on their surface (Figure 1(b^b^,e^b^)).

On the other hand, acetone and isopropanol are polar solvents with a relative permittivity of 20.7 and 17.9, respectively. Hence, both solvents were able to disperse SS particles with a certain degree of aggregation, resulting in surface roughness with a maximum RMS value of 21.80 nm (Table 3). In addition, the agglomeration in the isopropanol-added solution could be dispersed after it was shaken due to the like-dissolves-like concept. This dissolves-like concept indicates that solutes, which are PDMS, SS and epoxy in this work, will dissolve best in a solvent that has a similar chemical structure to them [65]. From Figure 3A-a, OH-bonds were found in isopropanol but not in hexane (Figure 3B-a) and acetone (Figure 3C-a) Therefore, PDMS, SS, and epoxy dissolved best in isopropanol instead of acetone or hexane.

### 3.4. FTIR of Different Types of Solvent and Coating Solution

FTIR spectroscopy was used to investigate functional groups in coating solutions of different types of solvent by analyzing their respective precursors. For isopropanol (Figure 3A-a), it can be seen that O-H stretching, and O-H bending are indicated at 3318 cm^−1^ and 951 cm^−1^, respectively. At 2970 cm^−1^, C-H stretching of CH_3_ group is indicated. In addition, C-H bending vibration of CH_2_ and CH_3_ are shown at 1467 cm^−1^ and 1379 cm^−1^, correspondingly. A peak noticed at 3338 cm^−1^ corresponds to O-H stretching, which is attributed to its hydroxyl-terminated structure of PDMS (Figure 3A-b). At 1640 cm^−1^, O-C = O stretching is detected. Peaks that are noticed at 1259 cm^−1,^ 1022 cm^−1^ and 795 cm^−1^ in Figure 3A-b, representing Si-CH_3_, O-Si-O and Si-C bonds, respectively [36,66,67]. The finding of functional groups in isopropanol and PDMS matches with their respective chemical formulae of C_3_H_8_O and (C_2_H_6_OSi)_n_. For Mixture B (Figure 3A-c), PDMS, AMPS and DBTL were added into isopropanol. The reaction occurred in the preparation of Mixture B can be proven by the reduced intensity of O-H stretching at 3325 cm^−1^ and presence of C-N at 1200 cm^−1^, as a result of the AMPS is grafted at the hydroxyl-end of PDMS [6]. C-H stretching and bending of CH_3_ groups and O-H bending that were originally absent in PDMS spectra, were observed at 2969 cm^−1^, 1378 cm^−1^ and 951 cm^−1^, respectively, ascertaining the formation of methanol (CH_3_OH) as side-product in the proposed mechanism (Figure 4). Figure 3A-d shows FTIR spectrum of SS, in which O-H, Si-O and O-Si-O bonds are detected at 3318 cm^−1^, 1261 cm^−1^ and 1081 cm^−1^, respectively. A peak observed at 1640 cm^−1^ belongs to O-C = O stretching. In Figure 3A-e, O-H stretching is also found in epoxy (DGEBA) at 3319 cm^−1^. C-H stretching and rocking vibrations are noticed at 2967 cm^−1^ and 794 cm^−1^. In addition, C-O-C stretching and C-O stretching of oxirane ring are also found at 1022 cm^−1^ and 1259 cm^−1^ [6]. Lastly, Figure 3A-f shows the FTIR spectrum of isopropanol-coating. Both O-H stretching, and O-H out-of-plane bending are detected at 3338 cm^−1^ and 951 cm^−1^, respectively. Besides, C-H stretching of CH_3_, C-H bending vibrations of CH_2_ and CH_3_ groups are also detected at 2970 cm^−1^, 1467 cm^−1^ and 1379 cm^−1^, correspondingly. At 1261 cm^−1^, Si-CH_3_ bond is indicated. Peaks that are observed at 1128 cm^−1^ and 815 cm^−1^ correspond to O-Si-O and Si-C bonds, respectively. From 3A-c to Figure 3A-f, intensity of O-H stretching at ~3300 cm^−1^ increases, which may be attributed to the reaction occurred between Mixture B, epoxy and SS. Other than this, a strong and sharp peak observed at 951 cm^−1^ corresponds to high intensity of O-H out-of-plane bending may be caused by the presence of methanol as side-product [66].

Next, Figure 3B shows FTIR spectra of hexane solution and its precursors. The functional groups of PDMS (Figure 3B-b), SS (Figure 3B-d) and epoxy (Figure 3B-e) and are similar to those in Figure 3A. The precursor was included in Figure 3B for comparison purpose. For hexane (Figure 3B-a), with chemical formula of C_6_H_14_, O-H bond is absent, while C-H stretching of CH_3_ (2925 cm^−1^), C-H bending of CH_2_ and CH_3_ groups (1467 cm^−1^ and 1379 cm^−1^), as well as C-C stretching (1260 cm^−1^) are detected. In this case, reaction between PDMS and AMPS is verified as O-H bonds from PDMS are fully contributed in this reaction, causing depletion of O-H bonds and formation of C-N groups as observed at 1218 cm^−1^ in Mixture B (Figure 3B-c). Based on Figure 3B-f, O-H bond is not detected after the addition of epoxy and SS, which may be due to the polarity of hexane that results in agglomeration of SS particles; thus, the solution is not well-dispersed [46].

FTIR spectra of acetone solution and its precursors are shown in Figure 3C. The functional groups of PDMS (Figure 3C-b), SS (Figure 3C-d) and epoxy (Figure 3C-e) are similar to those in Figure 3A. The precursor was included in Figure 3C for comparison purpose. For acetone (Figure 3C-a), shows peaks correspond to C = O (1711 cm^−1^), C-H stretching of CH_2_ (1423 cm^−1^) and CH_3_ groups (1361 cm^−1^), together with C-C stretching (1220 cm^−1^), which are identical to its chemical formula of C_3_H_6_O [68]. For Mixture B (Figure 3C-c) of acetone-added solution, the presence of C-N groups at 1200 cm^−1^ and O-H stretching is also absent as O-H groups from PDMS and are utilized for bonding between AMPS and PDMS. In acetone-added coating solution (Figure 3C-f), intensity of O-H stretching noticed at 3479 cm^−1^ is low. This may be attributed to “like-dissolves-like” concept, leading to a higher degree of dispersion as compared to hexane-added solution but lower than that of isopropanol-added solution, as acetone is a polar solvent, but hydroxyl group is absent [65].

As seen, compared to S1 solution, S2 solution has similar peaks expect for peaks that correspond to O-H stretching and bending vibration. When S3 solution compared S1, an extra peak of C=O was observed while the Si-CH_3_ and O-H bending vibration were absence. Absence of C-H stretching (~2900 cm^−1^) and Si-CH_3_ bond (~1260 cm^−1^) in S3 is probably due to oxidation of C-H to form C=O bonds [69,70]. The oxidation is further affirmed by color of acetone-added solution, which is brownish in color as compared to other solutions (Figure 2).

### 3.5. The Mechanism of Reaction of S3 Solution

Since S3 with isopropanol solvent shows the best hydrophobic behavior, the mechanism of this solution is proposed. A mechanism for reactions occurred in the preparation of isopropanol-added solution is illustrated schematically in Figure 4a–c, based on FTIR analysis. In the preparation of Mixture B (Figure 3A-c), Si-O-Si (refer A) bonds were formed upon the mixing of AMPS and PDMS with the presence of DBTL as catalyst, by hydrogen bonding with PDMS at the –OCH_3_ ends of AMPS and methanol is formed as by-product (Figure 4a). Then the ring opening reaction of epoxy occur with AMPS-PDMS at the amine end and forming modified PDMS (Figure 4b, refer B). Then, the modified PDMS is attached to a central Si atom of SS particles, by forming hydrogen bonding and covalent bond after heating and release water (Figure 4c, refer C).

### 3.6. Peel of Test

A peel-off test was performed on S3 coated tiles to determine the durability of the coating on the substrate. This test was repeated five times. In Figure 5, the appearance of the tape surface and the water contact angle after the peel off test are presented. As seen, the surface of the tape is clear from any debris, indicating the coating’s resistance to separation from the substrate. This characteristic implies that the coating has a good adhered to the surface and forms a strong bond. As a result, there are no significant changes in the topography and roughness of the S3 sample before and after peeling, resulting in a WCA that is identical.

## 4. Conclusions

In this work, the best superhydrophobic coating has been synthesized using PDMS, SS, and epoxy resin (DGEBA) with isopropanol as the solvent rather than hexane and acetone. Isopropanol, which is a polar solvent, poses dissolving-like behavior towards PDMS, SS, and epoxy and aids in the creation of multiscale roughness. The S3 coated sample has a water contact angle of 152 ± 2° after UV irradiation and showed good durability. Such a superhydrophobic coating can be utilized on tiles to act as a self-cleaning surface.

## Data Availability

Not applicable.

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
