# Peer review of "Effect of Solvent on Superhydrophobicity Behavior of Tiles Coated with Epoxy/PDMS/SS"

_polymers, 2022, doi:10.3390/polym14122406_

Round 1

Reviewer 1 Report

  1. In the introduction section, various methods have been used for preparing rough surface. Here are some related literature that offer another mathod, wchih may enrich the methods description (International Journal of Hydrogen Energy 46 (2021) 26489-26498;  Membranes 2022, 12, 222; Journal of Power Sources 525 (2022) 231121).
  2. The roughness of the hydrophobic surfaces were evaluated using AFM in Fig. 1. The question is the test capacity of AFM tip may ranges in hundred nanometers, but the surface of these sample seems more rough, is it suitable or accurate enough to test using AFM? As I know , the profile tester with several micrometer range may be more suitable.
  3. To be frank, it is difficult to figure out the difference between the images displayed in Fig. 5, SEM images may help. 
  4. Following the last question, the WCA seems to be same displayed in Fig.5.
  5. the timeless of this worjk should be enhanced by citing more papers published in recent five years.

Author Response

Dear Reviewer,

Thanks for the comment. The response to the reviewer is uploaded as PDF file.

Thank you

Srimala

Reviewer 2 Report

The authors present a study on the effect of solvents on the hydrophobicity behavior of coatings based on epoxy/polydimethylsiloxane/silica hybrid materials.

To judge from the quite well-written introduction section which comprises a vast literature revision, this manuscript aims to evaluate the effect of different solvents such as hexane, acetone, and isopropanol in the hydrophobicity of these epoxy/polydimethylsiloxane/silica coatings.

In my opinion, the manuscript can be accepted for publication only after a major revision of the following queries:

Line 214:

The name and model of the AFM instrument must be included in this paragraph.

Figure 1:

Were these AFM images subjected to background subtraction? Otherwise, the minor height in AFM line profiles should be normalized to zero.

Line 397 & 409:

It is not clear what the authors refer to with O-C-O stretching at 1640 cm-1. Is it the O-C=O stretching mode?

It is quite clear to me that the peak at 1640 cm-1 is attributed to H2O bending modes of humidity adsorbed to the silica surface.

Authors can refer to J. Phys. Chem. C 2014, 118, 2454−2462.

Figure 3:

FTIR spectra X-axis must be "wavenumber" instead of "wavelength". Also, the vibrational mode assignation in the Figure must be more precise, particularly for those Si-O and Si-C modes. In addition, why do the AMPS vibrational modes associated with C-N groups are not assigned in FT-IR spectra?

Figure 4:

The mechanism depicted in Figure 4 is quite speculative only based on FTIR data. The validity of this mechanism must be accompanied by a more complete and unambiguous assignation of vibrational modes in Figure 3 and the text.

Author Response

Dear Reviewer,

Thanks for the comment and the respond is uploaded as a PDF file for your perusal. 

Thanks

Srimala

Round 2

Reviewer 1 Report

1.       There are many format errors in the reference citation in the manuscript, please carefully check and revise. For example, the first, second, and last paragraph of introduction and section 2.5.2.

2.       The WCA inset in Fig.1 should be measured and marked in the corresponding image. And the data analysis of the WCA should be offered and correlated with the roughness analysis.

3.       “As seen, there are no remarkable changes in water contact angle for both samples, indicating strong adherence between coating and tiles substrate”, could the author explain in detail the relationship between the contact angle and the adherence of the coating on the substrate?

4.       “In this work, the best superhydrophobic coating has been synthesized using PDMS, SS, and epoxy resin (DGEBA) in isopropanol solvent.” It was declared in the conclusion that the best superhydrophobic coating was obtained, compared to which? How to define the best?

5.       The revisions corresponds to the comments should be marked in the updated version and displayed in the response point to point.

Author Response

Thank you very much for the comments and attached is the revision done for round 2. The changes has been highlighted in yellow in revised manuscript.

Regards

Srimala

Reviewer 2 Report

Authors have addressed my major concerns about the manuscript in its original form and, in my opinon, the revised manuscript can be accepted for publication.

Note that there are format errors in some citations in the text.

Author Response

Thank you very much for the comments and the citations has been corrected and highlighted in yellow in the revised manusacipt.

Regards

Sri 

Round 3

Reviewer 1 Report

The paper has been improved as suggested.